# Simulate Time-integrated Coarse-grained Molecular Dynamics with Geometric Machine Learning

**Xiang Fu**[*]
MIT CSAIL

**Tian Xie**[*]
MIT CSAIL

**Nathan J. Rebello**
Department of Chemical Engineering, MIT

**Bradley D. Olsen**
Department of Chemical Engineering, MIT

**Tommi Jaakkola**
MIT CSAIL

## Abstract

Molecular dynamics (MD) simulation is the workhorse of various scientific domains but is limited by high computational cost. Learning-based force fields have made major progress in accelerating ab-initio MD simulation but are still not fast enough for many real-world applications that require long-time MD simulation. In this paper, we adopt a different machine learning approach where we coarse-grain a physical system using graph clustering, and model the system evolution with a very large time-integration step using graph neural networks. A novel score-based GNN refinement module resolves the long standing challenge long-time simulation instability. Despite only trained with short MD trajectory data, our learned simulator can generalize to unseen novel systems, and simulate for much longer than the training trajectories. Properties requiring 10-100 ns level long-time dynamics can be accurately recovered at several-orders-of-magnitude higher speed than classical force fields. We demonstrate the effectiveness of our method on two realistic complex systems: (1) single-chain coarse-grained polymers in implicit solvent; (2) multi-component Li-ion polymer electrolyte systems.

## 1 Introduction

Molecular dynamics (MD) simulation techniques have become an essential computational tool in many natural science disciplines. A major limitation of MD simulation is its high computational cost. Past efforts on accelerating MD simulation (further explained in Appendix A) include Machine learning (ML) force fields (Unke et al., 2021b), coarse-grained (CG) models (Wang et al., 2019), enhanced sampling methods (Yang et al., 2019), etc. However, these methods are still limited by a time-integration at the femtosecond level, so millions of steps are needed to achieve a nanosecond level simulation. A large scale screening can still be extremely expensive or even infeasible. Learned simulators are also known to become unstable when simulated for a long time (Unke et al., 2021b).

In this paper, We present a different approach by learning a graph neural network (GNN) (Li et al., 2019b;a; Sanchez-Gonzalez et al., 2020; Pfaff et al., 2021) simulator, that: (1) coarse-grains a molecular system using graph clustering; (2) bypasses force computation and directly learns time-integrated acceleration from data. Each step of the learned simulator can be as long as $10^2 \sim 10^5$ steps in traditional MD. Despite the learned simulation is at a lower spatio-temporal resolution, important properties that we aim to extract from the original trajectories can still be preserved. A novel score-based (Song & Ermon, 2019; Shi et al., 2021) GNN is used to refine predicted structures and enables stable long-time simulation. Experiments show that our model can learn from limited amount of short training MD trajectories, and generate significantly longer MD simulation for novel systems to accurately obtain long-time dynamical properties, while achieveing several-orders-of-magnitude speedup compared to classical force fields.

---

[*]Equal contribution. Correspondence to: Xiang Fu at xiangfu@mit.edu

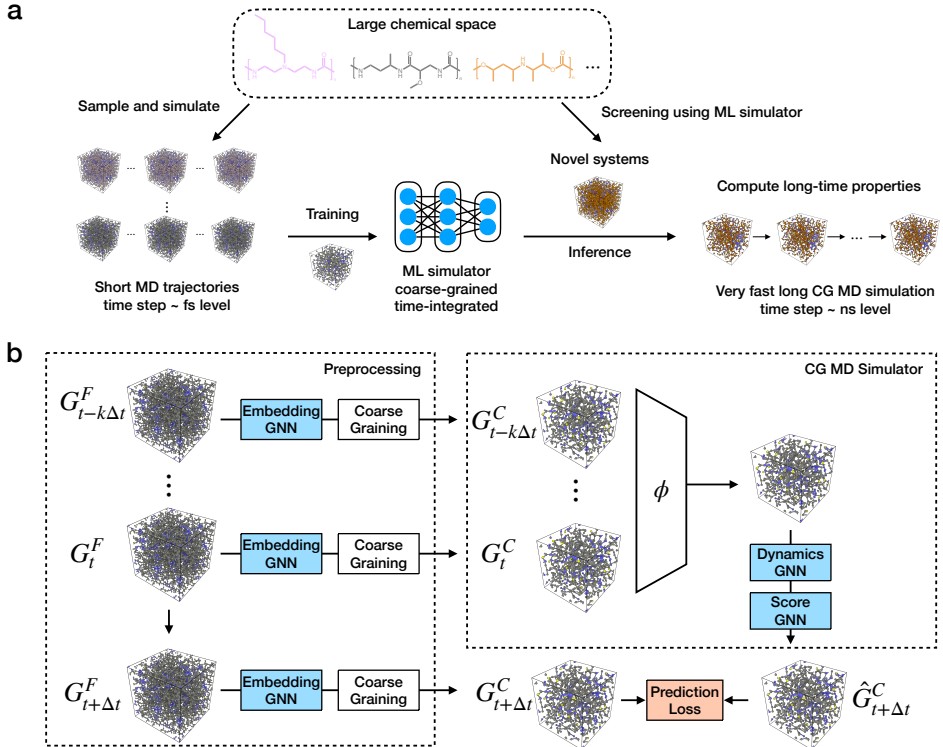

Figure 1: (a) To reduce the computational cost for estimating long-time properties of complex systems, our learned simulator is trained over short ground truth MD trajectories of sampled systems, and used to simulate long time-integrated CG MD trajectories for novel systems, while being order of magnitude more efficient. (b) Learning time-integrated CG MD with GNNs. Trainable modules are colored in blue. The loss term is colored in red. The preprocessing steps embed and coarse-grain an MD system to a coarse-level graph. The CG MD simulator featurize historical information using the featurizer $\phi$, after which the Dynamics GNN predict the next-step positions. A Score GNN can be optionally applied to further refine the predicted positions.

## 2 OVERVIEW OF PROPOSED METHOD

As shown in Figure 1 (a), we aim to learn a coarse-grained, time-integrated simulator from a list of short MD trajectories of atomic structures sampled from a given chemical space. Since the short MD trajectories cover most atomic structures and dynamics, we hope the simulator can generalize to novel atomic systems sampled from the same chemical space and run significantly longer simulations than the training trajectories. We can extract long-time properties from the longer trajectories which are hard to estimate with the short training trajectories.

The learned simulator predicts single-step time-integrated CG dynamics at time $t + \Delta t$ given the current CG state and $k$ historical CG states at $t, t - \Delta t, \ldots, t - k\Delta t$. Here $\Delta t$ is the time-integration step, which is significantly longer than that used in traditional MD simulation. Given ground truth trajectories at atomic resolution, we adopt a 3-step multi-scale modeling approach:

1. *Embedding*: learning atom embeddings at **fine level** using an Embedding GNN $\text{GN}_E$;
2. *Coarse-graining*: coarse-graining the system using graph clustering, and constructing CG-bead embedding from atom embedding learned at step 1;
3. *Dynamics* (and *Refinement*): learning time-integrated acceleration at **coarse level** using a Dynamics GNN $\text{GN}_D$. A Score GNN $\text{GN}_S$ is optionally learned to further refine the predicted structure.

All neural network modules in the pipeline are trained end-to-end to model single-step CG dynamics. To simulate new systems at test time, step 1 and 2 are the preprocessing steps, that are applied only

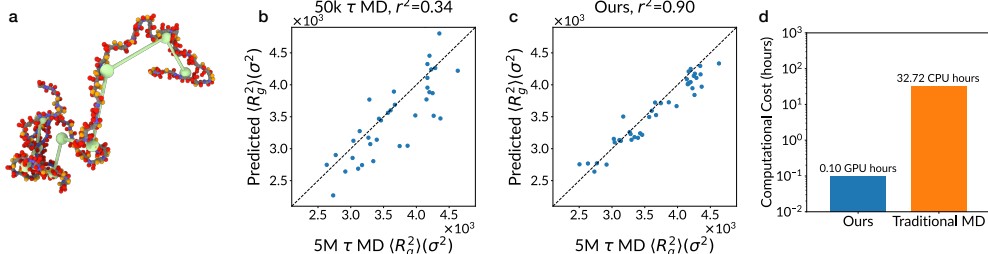

Figure 2: (a) Example polymer before and after coarse-graining. The green beads and bonds represent the CG structure. (b) Short MD of training trajectory length (50k $\tau$) gives high-variance, poor estimation of $\langle R_g^2 \rangle$. (c) $\langle R_g^2 \rangle$ estimation performance of our learned simulator. (d) Computational cost comparison of our learned simulation vs traditional MD. The y-axis is at log-scale.

once at the beginning to obtain an initial CG state. After preprocessing, step 3 is iteratively executed to simulate long MD trajectories. The generalizability of our model allows long-time properties of novel systems to be efficiently computed with significantly reduced computational cost. Figure 1 (b) provides an overview of our learned simulator. We explain the details of individual components and technical novelties in Appendix B.

## 3 HIGHLIGHTS OF EXPERIMENTAL RESULTS

We demonstrate the effectiveness of our approach by simulating two realistic complex systems of high practical significance: (1) single-chain coarse-grained polymers in implicit solvent; and (2) multi-component Li-ion polymer batteries. Both systems: (1) contain hundreds to thousands of particles and bonds; (2) involve bonded and non-bonded interactions between many different particle types. They are significantly more complex than previous applications of GNNs on physical simulation (Sanchez-Gonzalez et al., 2020).

**Single-chain coarse-grained polymers** are adopted from Webb et al. 2020. We train and test on different types of polymers, which poses great challenge on generalization. Our model coarse-grain every 100 beads into one CG-bead (Figure 2 (a)) and each time-integration step is 500 steps in traditional MD. We focus on the important property of radius of gyration ($R_g^2$). Figure 2 (b) shows the poor performance of using 50k $\tau$ (the time unit) short MD trajectories to estimate $\langle R_g^2 \rangle$ ($\langle \cdot \rangle$ denotes the mean operator) computed from 5M $\tau$ trajectories. More details on the dataset and experiment setting are included in Appendix C.

Table 1: Performance for predicting $R_g^2$ statistics. $r^2$ and MAE are computed for $\langle R_g^2 \rangle$, and EMD is computed for $R_g^2$ distribution. To evaluate EMD, the SL models output a Gaussian distribution with the predicted mean and variance.

| Method | $r^2$ (↑) | MAE (↓) | EMD (↓) |
|---|---|---|---|
| Short MD, 50k $\tau$ | 0.34 | 3.49 | 4.10 |
| GNN, SL | 0.70 | 2.63 | 2.82 |
| LSTM, SL | -0.91 | 5.59 | 5.82 |
| Ours, 5M $\tau$ | **0.90** | **1.40** | **1.60** |

$R_g^2$ **estimation using learned simulation.** We compare our model to two supervised-learning (SL) baseline models, one based on GNN and another based on long short-term memory (Hochreiter & Schmidhuber, 1997) (LSTM). SL models input the polymer graph structure, and directly predict mean and variance of $R_g^2$ without learning the dynamics (details in Appendix C). Table 1 summarizes various performance metrics of the two baseline models and our learned simulator. Figure 2 (c) shows the performance of our learned simulator in a regression plot. Our method significantly outperform the baseline models, and estimation from short 50k $\tau$ MD trajectories in all metrics. In particular, earth mover's distance (EMD) is a measure of distribution discrepancy. A lower EMD indicates a better match between the predicted $R_g^2$ distribution and the ground truth. Our learned simulator produces $R_g^2$ distributions that match well to the ground truth (more details in Appendix C).

With only short trajectories for training, The SL models can only fit to the high-variance statistics and produce poor results. Our testing polymers coming from a different distribution from training. We therefore observe a systemetic underestimation for the SL baseline (in Appendix C, Figure 4 (c, d)), and also high-variance prediction from the LSTM baseline. On the other hand, our learned simulator learns dynamics that generalize to a different class of polymers and longer time horizon,

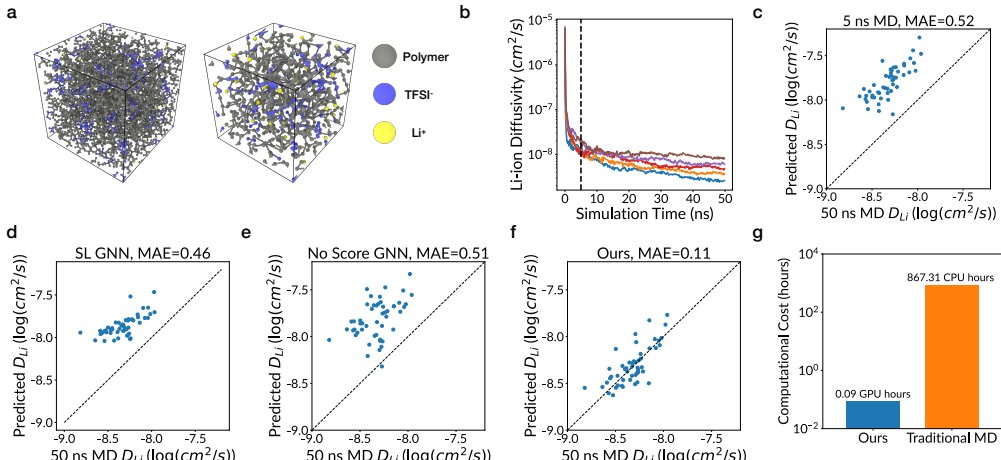

Figure 3: (a) Example SPE before and after coarse-graining. We color code the atoms by the molecules they belong to. (b) Slow convergence of Li-ion diffusivity for example SPEs. Estimating this long-time property with only 5 ns training trajectories (black dashed line) is very challenging. (c) 5-ns MD gives poor estimation of Li-ion diffusivity. (d,e,f) Performance of SL GNN model, learned simulator without Score GNN, and our full model in Li-ion diffusivity prediction. Only our full model is able to predict long-time property from short training trajectories. (g) Computational cost of our learned simulator vs traditional MD. The y-axis is at log-scale.

and produces accurate long-time property estimation. Our model is several orders of magnitude faster than tranditional MD (Figure 2 (d)). On a single RTX 2080 Ti GPU, our model takes only 0.1 GPU hours to finish a long simulation that traditional MD would take 32.7 CPU hours on a supercomputer. Furthermore, our model can also recover other highly dynamical properties such as the auto-correlation function, and relaxation time of $R_g^2$. We include these results in Appendix C.

**Multi-component Li-ion solid polymer electrolyte** (SPE, visualized in Figure 3 (a)) is a type of polymer system that is promising in advancing Li-ion battery technology (Zhou et al., 2019). We adopt the SPE systems introduced in Xie et al. 2021 where each MD trajectory contains a system with a distinct type of polymer mixed with lithium bis-trifluoromethyl sulfonimide (LiTFSI) under periodic boundary conditions. One key challenge in simulating the Li-ion transport in SPEs is the slow relaxation process in amorphous polymers. Consequently, estimating the diffusivity of particles like Li-ions requires very long simulation time to sample the dynamics (Figure 3 (b)). We aim to estimate particle diffusivity from short training trajectories only. We train on MD trajectories of 5-ns and evaluate on novel SPE trajectories of 50-ns long. On average each SPE system contains 6025 atoms. Our model groups every 7 bonded atoms into one CG-bead (Figure 3 (a)) and each time-integration step is 0.2 ns long, which is $10^5$ steps in traditional MD. More details on the dataset and experiment setting are included in Appendix D.

**Estimate long-time ion-transport properties using learned simulation.** Li-ion Diffusivity ($D_{\mathrm{Li}}$) computed from short MD significantly overestimates the the converged quantity (Figure 3 (c)). As only 5-ns trajectories are available for training, a strong SL GNN model adopted from Xie et al. (2021) that inputs the polymer chemical structure struggles to fix the systematic error (Figure 3 (d)), even after special treatment against overestimation (details in Appendix D). The novel Score GNN refinement module plays an important role in stablize long simulation. Without the Score GNN refinement, learned simulation becomes unphysical and overestimates $D_{\mathrm{Li}}$ (Figure 3 (e)). Our model with Score GNN refinement is able to learn accurate and generalizable dynamics, achieving an MAE of 0.11 for predicting 50-ns $D_{\mathrm{Li}}$ (Figure 3 (f)). One significant aspect of this result is long-time properties are accurately predicted with significantly shorter trajectoires for training. This is possible because the forces that govern the interaction between particles are consistent irrespective of simulation length, and our learned simulator is capable of learning the interaction from short trajectories. We summarize the diffusivity prediction performance for all particle types in Appendix D, where learned simulation produces the best result with a large margin. Figure 3 (g) compares the computational cost. Our model again uses orders-of-magnitude lower cost than traditional MD. We include more analysis, especially simulation stability analysis in Appendix D.

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

## A  RELATED WORKS

Machine learning (ML) force fields (Schoenholz & Cubuk, 2020; Doerr et al., 2021; Noé et al., 2020; Unke et al., 2021b) have made important progress towards accelerating MD simulations, achieving ab-initio level accuracy with significantly lower computational costs (Chmiela et al., 2017; 2018; Zhang et al., 2018; Jia et al., 2020) and generalizing to new atomic systems (Unke et al., 2021a; Park et al., 2021). However, learning-based force fields alone are still not enough to simulate complex atomic systems due to two major limitations: 1) the time integration step for the force fields are still at the femto-second level, which means that millions steps are needed to achieve a nano-second level simulation; 2) learning-based force fields are known to become unstable after long-time simulation (Unke et al., 2021b). Coarse-grained (CG) models (Wang & Gómez-Bombarelli, 2019; Wang et al., 2019; Husic et al., 2020) tackles these challenges by reducing system complexity and increase the time-integration steps. However, CG schemes are usually limited to specific systems (Noid, 2013) and still require short time-integration. Enhanced sampling methods offer a different approach (Bernardi et al., 2015; Yang et al., 2019; Wang et al., 2021) by modifying the potential energy surfaces to enable faster sampling of transition between metastable states. They often require identifying collective variables (CVs) in advance. ML has been used to augment these methods (Cendagorta et al., 2020) in free energy surface approximation (Schneider et al., 2017), obtaining latent CV (Sultan et al., 2018), etc. But it is still difficult to apply enhanced sampling methods for complex systems with large number of CVs (Wang et al., 2021). In addition, ML generative modeling approaches (Noé et al., 2019) has enabled fast sampling of equilibrium states across phases for systems, but are unable to produce realistic MD trajectories to study the dynamical behavior of atomic systems.

## B  METHOD DETAILS

High degree of coarse-graining and very long time integration greatly accelerate the simulation, but also make the dynamics non-Markovian and uncertain (Klippenstein et al., 2021). We introduce several technical modifications to tackle these challenges: (1) Historical information is used for predicting the next state; (2) The model predicts stochastic acceleration instead of deterministic forces; (3) A fine-level Embedding GNN constructs CG-bead types based on local atomic structures to characterize the coarse-level interactions; Model performance significantly drops without the Embedding GNN. (4) A Score GNN refinement step to resolve the long-standing instability problem of long-time learned simulation. Ablation study in Appendix F systematically demonstrates the influence of these modifications. Next, we explain each step of our learned simulator in details.

**Graph processing with graph neural networks.** A graph neural network takes graphs $G = (V, E)$ with node/edge features as inputs and processes a latent graph $G^h = (V^h, E^h)$ with latent node/edge representation through several layers of learned message passing. In this paper we adopt the EN-CODER-PROCESSOR-DECODER architecture (Sanchez-Gonzalez et al., 2020; Pfaff et al., 2021) for all GNNs, which inputs featurized graph and outputs a vector for each node. The ENCODER-PROCESSOR-DECODER model is constructed with three submodules, and a forward pass through the GNN follows three steps: (1) The ENCODER contains a node multi-layer perceptron (MLP) that is independently applied to each node, and an edge MLP that is independently applied to each edge to produce encoded node/edge features. (2) The PROCESSOR is composed of several layers of directed message passing layers. It generates a sequence of updated latent graphs and outputs the final latent graph. The message passing layers allows information to propagate between neighboring nodes/edges. (3) The DECODER is a node-wise MLP that is independently applied to the node features obtained from message passing to produce the outputs of a specified dimensionality. We refer interested readers to (Sanchez-Gonzalez et al., 2020) for more details on the GNN architecture.

In our implementation we use 2 hidden layers for all MLPs and 7 message-passing layers for all GNNs. The Embedding GNN $GN_E$ has a hidden size of 64, while the Dynamics and Score GNNs have a hidden size of 128. All activation functions in the neural networks are rectified linear unit (ReLU). We train the model for 2 million steps. The network is optimized with an Adam optimizer with an initial learning rate of $2 \times 10^{-4}$, exponentially decayed to $2 \times 10^{-5}$ over the 2 million training steps. All models are trained and used for producing long simulation over a single RTX 2080 Ti GPU.

**Representing MD trajectories as time series of graphs.** A ground truth MD simulation trajectory is represented as a time series of fine-level graphs $\{G_t^F\}$. The fine-level graph $G_t^F$ represents the MD state at time step $t$, and is defined as a tuple of nodes and edges $G_t^F = (V_t^F, E^F)$. Each node $\boldsymbol{v}_{i,t}^F \in V_t^F$ represents an atom [1], and each edge $\boldsymbol{e}_{i,j}^F \in E^F$ represents a chemical bond between the particles $\boldsymbol{v}_i^F$ and $\boldsymbol{v}_j^F$. The static fine-level graph $G^F = (V^F, E^F)$ describes all persistent features in a MD simulation, which include atom types, atom weights and bond types. These persistent features are used to construct a time-invariant node representation, that will be later used for CG-bead embedding in our model. Applying the CG model to the fine-level graphs $\{G_t^F\}$ produces the time series of coarse-level graphs $\{G_t^C\}$, where the CG state $G_t^C$ is defined by the tuple $G_t^C = (V_t^C, E_t^C)$. Each node $\boldsymbol{v}_{m,t}^C \in V_t^C$ represents a CG-bead, and each edge $\boldsymbol{e}_{m,n}^C \in E_t^C$ models an interaction between the CG-beads $\boldsymbol{v}_{m,t}^C$ and $\boldsymbol{v}_{n,t}^C$. Since both non-bonded interactions and bonded interactions at the coarse-level are significant towards dynamics modeling, the edge set $E_t^C$ contains both CG level bonds and radius cut-off edges constructed from the CG-bead coordinates at time $t$.

**Learning CG-bead type embeddings with Embedding GNN.** Each node in the static fine-level graph $\boldsymbol{v}_i^F = [\boldsymbol{a}_i^F, w_i^F]$ is represented with (1) a learnable type embedding $\boldsymbol{a}_i^F$ that is fixed for a given atomic number and (2) a scalar weight $w_i^F$ of the particle. The edge embedding is the sum of the type embedding of the two endpoints and a learnable bond type embedding: $\boldsymbol{e}_{i,j}^F = [\boldsymbol{a}_i^F + \boldsymbol{a}_j^F + \boldsymbol{a}_{i,j}^F]$. The bond type embedding $\boldsymbol{a}_{i,j}^F$ is also fixed for a given bond type. We input this graph $G^F$ to the Embedding GNN $\mathrm{GN}_E$ that outputs node embeddings $\boldsymbol{v}_i^F = [\boldsymbol{c}_i^F]$, for all $\boldsymbol{v}_i^F \in V^F$. This learned node embedding contains no positional information and will be used in the next coarse-graining step for representing CG-bead types. The Embedding GNN is trained end-to-end with Dynamics GNN and Score GNN.

**Graph-clustering CG model.** Our CG model assigns atoms to different atom groups using a graph clustering algorithm (e.g. METIS Karypis & Kumar (1998)) over the fine-level graph. The METIS clustering algorithm partitions atoms into groups of the similar size, while minimizing the number of chemical bonds between atoms in different groups. Each node in the fine-level graph is assigned a group number: $C(\boldsymbol{v}_{i,t}^F) \in \{1, \ldots, M\}$, for all $\boldsymbol{v}_{i,t}^F \in V_t^F$. Here $M$ is the number of atom groups (CG-beads).

**Construct CG graph states.** We represent each fine-level node $\boldsymbol{v}_{i,t}^F \in V_t^F$ with $\boldsymbol{v}_{i,t}^F = [\boldsymbol{c}_i^F, w_i^F, \boldsymbol{x}_{i,t}^F]$, where $\boldsymbol{c}_i^F$ is the learned node type embedding from $\mathrm{GN}_E$, $w_i^F$ is the weight, and $\boldsymbol{x}_{i,t}^F$ is the position. We can then obtain the coarse-level graph $G_t^C = (V_t^C, E_t^C)$ through grouping atoms in the same group into a CG-bead. Denote the set of fine-level atoms with group number $m$ as $C_m = \{i : C(\boldsymbol{v}_{i,t}^F) = m\}$. The representation of the CG-bead $\boldsymbol{v}_{m,t}^C = [\boldsymbol{c}_m^C, w_m^C, \boldsymbol{x}_{m,t}^C]$ is defined as following:

$$\boldsymbol{c}_m^C = \underset{C_m}{\mathrm{mean}}(\boldsymbol{c}_i^F) \equiv \frac{\sum_{i \in C_m} \boldsymbol{c}_i^F}{|C_m|}$$

$$w_m^C = \underset{C_m}{\mathrm{sum}}(w_i^F) \equiv \sum_{i \in C_m} w_i^F \tag{1}$$

$$\boldsymbol{x}_{m,t}^C = \underset{C_m}{\mathrm{CoM}}(\boldsymbol{q}_{i,t}^F, w_i^F) \equiv \frac{\sum_{i \in C_m} w_i^F \boldsymbol{x}_{i,t}^F}{\sum_{i \in C_m} w_i^F}$$

The operators $\mathrm{mean}_{C_m}$ stands for taking the mean over $C_m$, $\mathrm{sum}_{C_m}$ stands for taking the sum over $C_m$, and $\mathrm{CoM}_{C_m}$ stands for taking the center of mass over $C_m$. Applying this grouping procedure for all atom groups $m \in \{1, \ldots, M\}$ creates the set of all CG nodes $V_t^C = \{\boldsymbol{v}_{m,t}^C : m \in \{1, \ldots, M\}\}$ for the coarse-level graph $G_t^C = (V_t^C, E_t^C)$.

We next construct the edges $E_t^C$. *CG-bonds* are created for bonded atoms separated in different groups. We create a CG-bond $e_{m,n,t}^C$ if there exists a chemical bond between a pair of atoms in group $C_m$ and group $C_n$. That is:

$$e_{m,n,t}^C \in E_t^C \impliedby \exists i \in C_m, j \in C_n, \text{ such that } e_{i,j}^F \in E^F \tag{2}$$

---

[1] If the ground truth MD simulation already uses a CG model, each node will correspond to a CG-bead defined by the CG model, and correspondingly the set of edges will be the set of all CG chemical bonds. For the ease of presentation, in the rest of this paper we refer to particles in the fine-level graph as "atoms".

We further create radius cut-off edges by, for each CG-bead, finding all neighboring beads within a pre-defined connectivity raidus $r$ (with consideration to the simulation setup, e.g., periodic boundaries):

$$e_{m,n,t}^C \in E_t^C \impliedby \|\boldsymbol{x}_{m,t}^C - \boldsymbol{x}_{n,t}^C\| < r \tag{3}$$

We set a large enough connecitivity radius to capture significant interactions between all pairs of CG-beads.

**Learning CG MD with Dynamics GNN.** The Dynamics GNN $\mathrm{GN}_D$ inputs a featurized coarse graph state $\phi(\{G_{t-i\Delta t}^C\}_{i=0}^k)$ and outputs a distribution of the time-averaged acceleration for each CG-bead. The input node features $\boldsymbol{v}_{m,t}^C = [\boldsymbol{c}_m^C, w_m^C, \{\dot{\boldsymbol{x}}_{m,t-i\Delta t}^C\}_{i=0}^{k-1}]$ include the CG-bead type embedding $\boldsymbol{c}_m^C$, weight $w_m^C$, current and $k$-step history velocities $\{\dot{\boldsymbol{x}}_{m,t-i\Delta t}^C\}_{i=0}^{k-1}$. The input edge features $\boldsymbol{e}_{m,n,t}^C = [\boldsymbol{x}_{m,t}^C - \boldsymbol{x}_{n,t}^C, \|\boldsymbol{x}_{m,t}^C - \boldsymbol{x}_{n,t}^C\|, \boldsymbol{c}_{m,n}^C]$ include displacement and distance between the two end points, and an embedding vector $\boldsymbol{c}_{m,n}^C$ indicating whether $e_{m,n,t}^C$ is a CG-bond or is constructed through radius cut-off. The output is a 3-dimensional Gaussian: $\mathrm{GN}_D(\phi(\{G_{t-i\Delta t}^C\}_{i=0}^k)) = \mathcal{N}(\boldsymbol{\mu}_t, \boldsymbol{\sigma}_t^2) = \{\mathcal{N}(\boldsymbol{\mu}_{m,t}, \boldsymbol{\sigma}_{m,t}^2) : \boldsymbol{v}_{m,t}^C \in V_t^C\}$ [2], where $\boldsymbol{\mu}_t$ and $\boldsymbol{\sigma}_t^2$ are the predicted mean and variance at time $t$, respectively. The training loss $L_{\mathrm{dyn}}$ for predicting the forward dynamics is thus the negative log likelihood of the ground truth acceleration:

$$L_{\mathrm{dyn}} = -\log \mathcal{N}(\ddot{\boldsymbol{x}}_t^C | \boldsymbol{\mu}_t, \boldsymbol{\sigma}_t^2) \tag{4}$$

The end-to-end training minimizes $L_{\mathrm{dyn}}$, which is only based on single-step prediction of time-integrated acceleration. At inference time (for long simulation), the predicted acceleration $\hat{\ddot{\boldsymbol{x}}}_t \sim \mathcal{N}(\boldsymbol{\mu}_t, \boldsymbol{\sigma}_t^2)$ is sampled from the predicted Gaussian and integrated with a semi-implicit Euler integration to update the positions to the predicted positions $\hat{\boldsymbol{x}}_{t+\Delta t}$:

$$\begin{aligned}
\hat{\dot{\boldsymbol{x}}}_{t+\Delta t} &= (\dot{\boldsymbol{x}}_t + \hat{\ddot{\boldsymbol{x}}}_t \Delta t) \\
\hat{\boldsymbol{x}}_{t+\Delta t} &= \boldsymbol{x}_t + \hat{\dot{\boldsymbol{x}}}_{t+\Delta t} \Delta t
\end{aligned} \tag{5}$$

**Learning to refine CGMD predictions with Score GNN.** We introduce the Score GNN, a score-based generative model (Song & Ermon, 2019) to resolve the stability issue of long simulation for complex system. Following the noise conditional score network (NCSN) framework (Song & Ermon, 2019; Shi et al., 2021), the Score GNN is trained to output the gradients of the history-conditional log density (i.e. scores) given history state information and CG-bead coordinates as input. An incorrect 3D configuration can be refined through iteratively applying the learned scores. With the refinement step, each forward simulation step follows a predict-then-refine procedure. The Dynamics GNN first predict the (potentially erroneous) next-step positions, which the Score GNN refine to the final prediction $\hat{\boldsymbol{x}}_{t+\Delta t}$.

During training, the Score GNN is trained to denoise noisy CG-bead positions to the correct positions. Let the noise levels be a sequence of positive scalars $\sigma_1, \dots, \sigma_L$ satisfying $\sigma_1/\sigma_2 = \cdots = \sigma_{L-1}/\sigma_L > 1$. The noisy positions $\tilde{\boldsymbol{x}}_{t+\Delta t}^C$ is obtained by perturbing the ground truth positions with Gaussian noise: $\tilde{\boldsymbol{x}}_{t+\Delta t}^C = \boldsymbol{x}_{t+\Delta t}^C + \mathcal{N}(0, \sigma^2)$, where multiple levels of noise $\sigma \in \{\sigma_i\}_{i=1}^L$ are used. Due to coarse-graining, the coarse-level dynamics is non-Markovian (Klippenstein et al., 2021). Given the current state and historical information $\mathcal{H}$ that causes $\boldsymbol{x}_{t+\Delta t}^C$, the noisy positions follow the distribution with density:

$$p_\sigma(\tilde{\boldsymbol{x}}_{t+\Delta t}^C | \mathcal{H}) = \int p_{\mathrm{data}}(\boldsymbol{x}_{t+\Delta t}^C | \mathcal{H}) \mathcal{N}(\tilde{\boldsymbol{x}}_{t+\Delta t}^C | \boldsymbol{x}_{t+\Delta t}^C, \sigma^2 I) d\boldsymbol{x}_{t+\Delta t}^C$$

The Score GNN outputs the gradients of the log density of particle coordinates that denoise the noisy particle positions $\tilde{\boldsymbol{x}}_{t+\Delta t}^C$ to the ground truth positions $\boldsymbol{x}_{t+\Delta t}^C$, conditional on the current state and historical information. That is: $\forall \sigma \in \{\sigma_i\}_{i=1}^L$,

---

[2]for ease of notation, in the rest of this paper we omit the node/edge indices when referring to every node/edge in the graph

$$\begin{aligned}
\text{GN}_S([\boldsymbol{v}_t^{Ch}, \tilde{\boldsymbol{x}}_{t+\Delta t}^C]) &= \nabla_{\tilde{\boldsymbol{x}}_{t+\Delta t}^C} \log p_\sigma(\tilde{\boldsymbol{x}}_{t+\Delta t}^C | \boldsymbol{v}_t^{Ch})/\sigma \\
&\approx \nabla_{\tilde{\boldsymbol{x}}_{t+\Delta t}^C} \log p_\sigma(\tilde{\boldsymbol{x}}_{t+\Delta t}^C | \mathcal{H})/\sigma \qquad\qquad (*) \\
&= \mathbb{E}_{p(\boldsymbol{x}_{t+\Delta t}^C | \mathcal{H})}[(\boldsymbol{x}_{t+\Delta t}^C - \tilde{\boldsymbol{x}}_{t+\Delta t}^C)/\sigma^2]
\end{aligned}$$

Note that in step $(*)$ we are approximating $\mathcal{H}$ with learned latent node embeddings $\boldsymbol{v}_t^{Ch}$, which is the last-layer hidden representation output by the Dynamics GNN. $\boldsymbol{v}_t^{Ch}$ contains current state and historical information. The training loss for $\text{GN}_S$ is presented in Equation (6), where $\lambda(\sigma_i) = \sigma_i^2$ is a weighting coefficient that balance the losses at different noise levels. During training, the first expectation in Equation (6) is obtained by sampling training data from the empirical distribution, and the second expectation is obtained by sampling the Gaussian noise at different levels.

With the Score GNN, training is still end-to-end by jointly optimizing the dynamics loss and score loss: $L = L_{\text{dyn}} + L_{\text{score}}$. At inference time, the refinement starts from the predicted positions from Dynamics GNN. We use annealed Langevin dynamics, which is commonly used in previous work (Song & Ermon, 2019), to iteratively apply the learned scores to gradually refine the particle positions, with a decreasing noise term. The predict-then-refine procedure can be repeated to simulate complex systems for long time horizon stably.

$$L_{\text{score}} = \frac{1}{2L} \sum_{i=1}^{L} \lambda(\sigma_i) \mathbb{E}_{\text{data}(\boldsymbol{x}_{t+\Delta t}^C | \mathcal{H})} \mathbb{E}_{p_{\sigma_i}(\tilde{\boldsymbol{x}}_{t+\Delta t}^C | \boldsymbol{x}_{t+\Delta t}^C)} \left[ \left\| \frac{\text{GN}_S([\boldsymbol{v}_t^{Ch}, \tilde{\boldsymbol{x}}_{t+\Delta t}^C])}{\sigma_i} - \frac{\boldsymbol{x}_{t+\Delta t}^C - \tilde{\boldsymbol{x}}_{t+\Delta t}^C}{\sigma_i^2} \right\|_2^2 \right] \qquad (6)$$

## C    EXPERIMENT DETAILS: SINGLE-CHAIN CG POLYMERS IN IMPLICIT SOLVENT

Polymer functionality is decided by macromolecular properties at a large time/length-scale, and (CG) MD simulation has been widely used to obtain important properties in practical material design efforts (de Pablo, 2011; Shmilovich et al., 2020; Webb et al., 2020). We adopt the polymers introduced in (Webb et al., 2020), where all polymers are composed of four types of beads and ten types of constitutional units (CUs). We train on regular copolymers with a repeat pattern of four CUs (class-I), and test on random polymers constructed from four CUs (class-II). This distribution shift (Figure 4 (a)) poses great challenge on generalization. All polymers are randomly sampled and simulated under LJ units with a time-integration of $0.01 \tau$. Each polymer on average contains 889.5 beads. We train all models on 100 short class-I MD trajectories (with 10 trajectories for validation) of 50k $\tau$, and evaluate on 40 testing class-II polymers using trajectories of 5M $\tau$ (100x longer). The single chain polymers are simulated using LAMMPS (Thompson et al., 2022), with the force-field parameters and chemical space defined in (Webb et al., 2020). We refer interested readers to (Webb et al., 2020) for more details on the simulation set up.

Radius of gyration ($R_g^2$) is a property practically related to rheological behavior of polymers in solution and polymer compactness that are useful for polymer design (Altintas & Barner-Kowollik, 2016; Upadhya et al., 2019; Webb et al., 2020). Reliable estimation of $R_g^2$ statistics requires sampling with long enough MD simulation, while short simulation result in high-variance and significant error. Figure 2 (b) shows how $R_g^2$ rapidly changes with time.

As we use very aggressive CG modeling with every 100 beads into a super CG-bead, We use another neural network that inputs the coarse-level latent graph representation to fit the residual of $R_g^2$ caused by this high degree of coarse-graining. A time integration of $\Delta t = 5\tau$ is used, making every step of the learned simulator models the integrated dynamics of 500 steps from training trajectory. At test time, we use the learned simulator to generate 5M $\tau$ trajectories and compute properties. With the system significantly simplified by coarse-graining, long and stable simulation can be achieved without using a Score GNN for refinement. A refinement module does not improve nor deteriorate performance, but would slow down the learned simulation.

**Single-chain CG polymer dataset.** The single chain polymers are simulated using LAMMPS (Thompson et al., 2022), with the force-field parameters and chemical space defined in (Webb et al., 2020). All simulations are done in reduced units with characteristic quantities $\sigma$ for distance and $\tau$

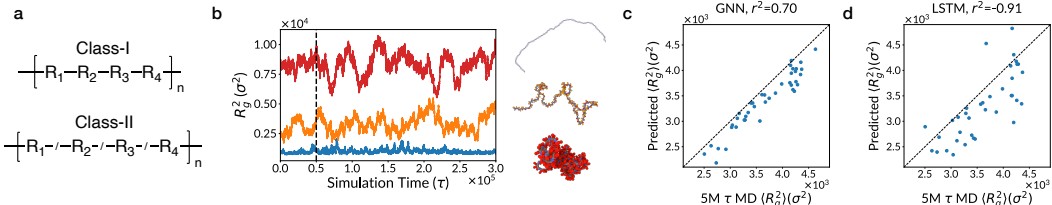

Figure 4: (a) Class-I polymers are used for training, while class-II polymers are used testing. The structure variation requires the model to learn generalizable dynamics. (b) $R_g^2$ for the three training polymers with smallest, median, and largest $\langle R_g^2 \rangle$, over a 300k $\tau$ period. Our training trajectories are 50k $\tau$ long (black dashed line), while we use 5M $\tau$ long trajectories for evaluation. (c, d) $\langle R_g^2 \rangle$ estimation performance of the supervised learning baselines using (c) GNN and (d) LSTM.

for time. Single-chain CG polymer dynamics in implicit solvent evolves according to the Langevin equation using the velocity-Verlet integration scheme. We use a time step of $0.01\tau$. Training trajectories are 50k $\tau$ after removing the initial trajectory for relaxation, and are recorded every 5 $\tau$. Testing trajectories are 5M $\tau$, and are recorded every 500 $\tau$. The polymer interaction is described by summation of bonded and non-bonded potential energy functions. We refer interested readers to (Webb et al., 2020) for more details on the simulation set up.

**Calculation of single-chain polymer properties.** Our main property of study, squared radius of gyration ($R_g^2$) is computed by:

$$R_g^2 = \left( \frac{\sum_{i \in V} m_i d_i^2}{\sum_{i \in V} m_i} \right)$$

where $V$ is the set of all nodes, $m_i$ is the mass of particle $i$, and $d_i$ is the distance from particle $i$ to the center of mass.

The relaxation time for $R_g^2$ is derived from the autocorrelation function (ACF), which is computed as:

$$\text{ACF}(y) = \frac{\langle R_g^2(t)^2 R_g^2(t+y)^2 \rangle_t - (\langle R_g^2(t)^2 \rangle_t)^2}{\langle R_g^2(t)^4 \rangle_t - (\langle R_g^2(t)^2 \rangle_t)^2}$$

Here $\langle \cdot \rangle_t$ stands for averaging over the entire trajectory. The relaxation time $t_{R_g^2}$ is computed by fitting an exponential function $f(y) = \exp(-t_{R_g^2} y)$ to the ACF, and the relaxation time is the time when the ACF decays to $1/e$. It is a highly dynamical long-time property decided by the polymer structure.

**Single-chain polymer baseline models.** We conduct experiments with two supervised learning baseline models. The first one is a GNN model that takes the polymer chemical graph as input and output the mean and standard deviation of $R_g^2$. Each node in the polymer graph is a CG-bead and each edge is a chemical bond. The GNN uses the ENCODER-PROCESSOR-DECODER with 10 message-passing layers to process the polymer graph to a latent graph with node/edge embeddings. The MLPs in the GNNs has 2 hiden layers with size 128. The node embeddings are then summed to get the graph embedding, which is then processed by a 2-layer MLP with hidden size 256 to obtain the final prediction of mean and standard deviation of polymer $R_g^2$. The second LSTM baseline model takes the 1D polymer chain structure as its input, and replaces the GNN encoder with an 2-layer LSTM encoder with hidden size 256. All activation functions in the neural networks are ReLU. The baseline models are optimized with an Adam optimizer with a learning rate of $10^{-3}$, exponentially decayed to $5 \times 10^{-4}$ over 1500 training epochs. Due to the limited time horizon of training data, the baseline models can only fit to high-variance labels, leading to underperforming prediction results.

**Recovery of long-time distributional and dynamical properties.** For the testing polymer with median $\langle R_g^2 \rangle$, Figure 5 (a) demonstrates that the $R_g^2$ distribution produced by our model matches very well with the ground truth, confirming the low EMD achieved. Furthermore, we show our learned simulator can capture the long-time dynamics in the autocorrelation function (ACF) of $R_g^2$. Figure 5 (b) shows the ACF computed from our learned simulation matches the ground truth. We

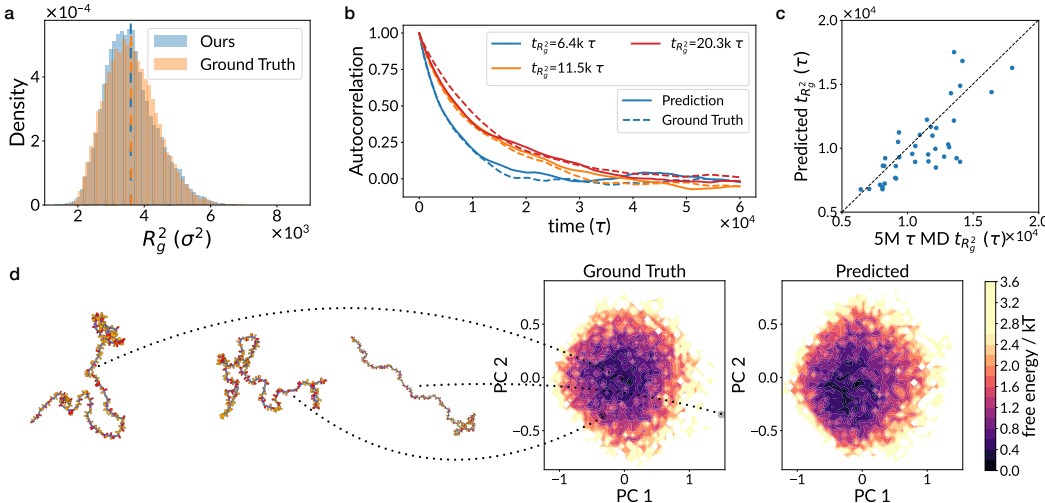

Figure 5: (a) $R_g^2$ distribution computed from our learned simulation matches the ground truth. (b) The autocorrelation function of $R_g^2$, for the three polymers with the smallest, median, and the largest relaxation time. (c) Prediction performance of our model on the $R_g^2$ relaxation time. (d) Two dimensional free energy surface produced with PCA, with representative states with low/high free energy visualized.

obtain the relaxation time $t_{R_g^2}$ from the ACFs (details in Method) and achieve a $r^2$ of $0.48$ (Figure 5 (c)) using learned simulation. The relaxation time is a long-time dynamical property that is significantly more challenging to estimate than $\langle R_g^2 \rangle$. It can not be obtained from many independent samples of polymer states, and requires simulation much longer than our training trajectories to estimate accurately. Recovery of $t_{R_g^2}$ shows that our model captures realistic dynamics rather than just the distribution of states. Finally, we also visualize 2-dimensional free energy surfaces of the ground truth simulation and our learned simulation in Figure 5 (d). The surfaces are produced by fitting a projection map using principle component analysis (PCA) over the ground truth CG-bead pairwise distances, and apply the same projection to the model-predicted data.

## D EXPERIMENT DETAILS: MULTI-COMPONENT LI-ION SOLID POLYMER ELECTROLYTE SYSTEMS

Atomic-scale MD simulations has been an important tool in SPE studies (Webb et al., 2015; Savoie et al., 2017; Wang et al., 2020; Xie et al., 2021), but are very computationally expensive. Usually containing multiple components with thousands of atoms, the SPE systems are inherently very complex. The challenge also comes from their amorphous nature and the slow convergence of key quantities, requiring long MD simulations on the order of 10 to 100 ns to estimate. In our experiments, we train on 530 short MD trajectories of 5 ns (with 30 trajectories for validation), and evaluate on 50 novel SPE trajectories of 50-ns. The SPE systems are simulated using LAMMPS (Thompson et al., 2022) with the force-field parameters and chemical space defined in (Xie et al., 2021). We refer interested readers to (Xie et al., 2021) for more details on the simulation set up. Figure 6 (a) shows the chemcial space of the polymer in Xie et al. 2021. Figure 6 (b) shows the performance comparison on particle diffusivity of all types.

The SPE systems are simulated using LAMMPS (Thompson et al., 2022) with the force-field parameters and chemical space defined in (Xie et al., 2021). For all systems, there are 50 Li-ions and TFSI-ions in the simulation box and each polymer chain has 150 atoms in the backbone. The training trajectories are 5-ns long after removing initial equilibration, while the testing trajectories are 50-ns long. Each system is run in the canonical ensemble (nVT) at a tempreature of 353K using a multi-timescale integrator with an outer timestep of 2 fs for non-bonded interactions, and an inner timestep of 0.5 fs. Both training and testing trajectories are recorded every 2 ps. The atomic interac-

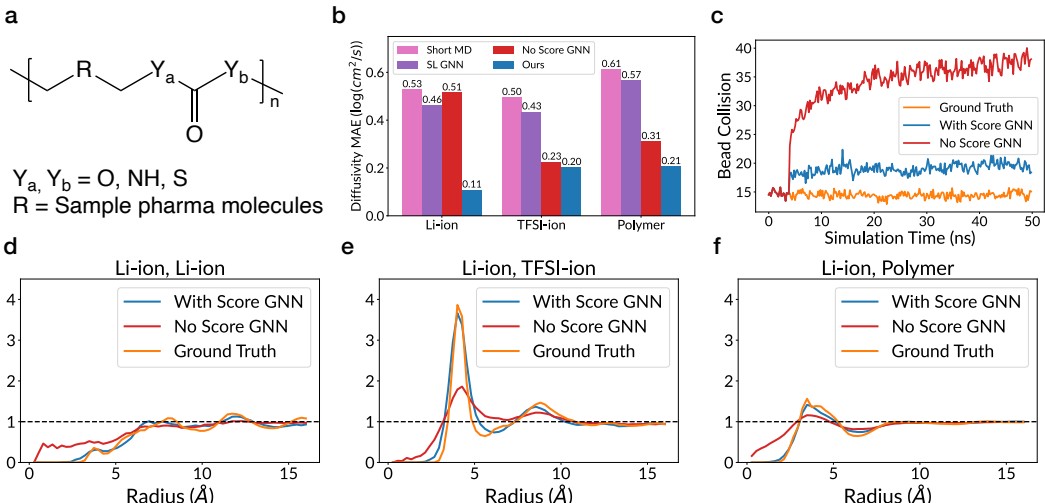

Figure 6: (a) The chemical space for the polymer in SPEs. (b) Performance of all models in predicting diffusivity of Li-ion, TFSI-ion, and polymer particles. (c) Bead collision as a function of simulation time, averaged over the 50 testing SPEs for all methods. (d) RDF of Li-ions, for our model with/without the Score GNN refinement, and the ground truth MD simulation, averaged over a 50-ns trajectory of an example SPE. (e) RDF of Li-ions and TFSI-ions. (f) RDF of Li-ions and polymer particles.

tions are described by the polymer consistent force-field (PCFF+) (Sun, 1994; Rigby et al., 1997). We refer interested readers to (Xie et al., 2021) for more details on the simulation set up.

**Calculation of SPE properties.** Ion transport properties of SPEs is the topic of study for many previous research, and requires long-time simulation to estimate. In particular, our experiments focus on particle diffusivity. With a trajectory of time horizon $T$, the diffusivity $D$ of a particle is computed by:

$$D = \frac{\|x_T - x_0\|_2^2}{T}$$

where $x_T$ is the position at time $T$, and $x_0$ is the position at time $0$. The diffusivity of a type of particle (e.g., Li-ion) is then computed by averaging the particle diffusivity over all particles of that type.

The radial distribution function (RDF) describes the particle density as a function of distance from a reference particle. For a system with many distinct types particles, we can compute RDF for particular types by counting the neighbor atoms of certain types at various radius and each time step, and average over the entire trajectory. The RDF for particle types $A, B$ at distance $r$ is computed as:

$$\text{RDF}_{A,B}(r) = \frac{d[n_{A,B}(r)]}{4\pi r^2 dr}$$

Here $n_{A,B}(r)$ is the number of particle pairs with types $A$ and $B$ and distance in $[r, r + dr)$, where $dr$ is a small bin size.

**SPE diffusivity prediction baseline model.** We adopt the GNN model proposed in (Xie et al., 2021) and refer interested readers to (Xie et al., 2021) for more details. The only modification we make is changing the training objective, so as to recitfy the overestimation coming from short MD horizon of the training data. We let the baseline model fit the 5 ns diffusvity curves by predicting two parameters in the function: $f(t) = y + e^{-\theta t}$. Therefore, the model approximates the converging process of diffusivity with exponenial decay. During evaluation, we set $t = 50$ ns to obtain the model prediction for 50-ns diffusivity. However, without any long training trajectories, it is very challenging to guess the decaying process of diffusivity for various SPEs. The baseline model performs better than using 5-ns ground truth MD, but still significantly overestimate particle diffusivity.

**Stability of learned simulation and Score GNN refinement.** Unlike the single-chain polymers, learned simulation for the more complex SPEs suffers from the out-of-distribution prediction prob-

lem, which we resolve with the Score GNN refinement step. As a measure of stability, we consider a "bead collision" happens when two beads have a distance below 1.0 Å. A large number of bead collision indicates the system is unphysical. Figure 6 (c) demonstrates the number of bead collison as a function of simulation time, averaged over all 50 test SPEs. Simulation without Score GNN refinement suffers from bead collision, and the system becomes increasingly unphysical as simulation proceeds. The Score GNN refinement significantly reduces collision. The system remains stable and realistic for the entire 50-ns simulation. In Figure 6 (d), (e), and (f) we plot radial distribution functions (RDFs) averaged over the 50-ns simulation for an example testing SPE over (d) Li-ions; (e) Li-ions and TFSI-ions; and (f) Li-ions and polymer beads. The excessive high density at near distance and missed radius peaks confirm the bead collision and instability without Score GNN refinement. With Score GNN refinement, the learned simulation produces RDFs that can recover the overall shape and radius peaks of the ground truth. We include more RDF plots in Figure 11.

## E  ABLATION STUDY

We conduct ablation experiments to study the influence of our modeling choices, using the SPE dataset.

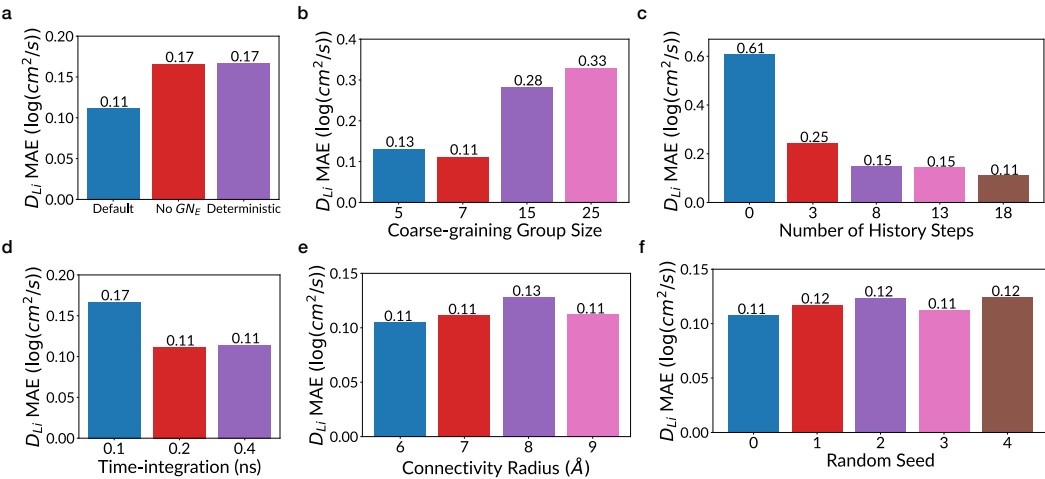

Figure 7: (a) Comaprison of our default model, a model without the Embedding GNN at the fine-level graph, and a model that predicts deterministic acceleration for each particle. (b) Comparison of different coarse-graining atom group size. The atom group size is the number of atoms contained in each CG-bead. Performance drops when the coarse-graining is too fine (5) or too coarse(15, 25). (c) Comparison of different history length for predicting the dynamics. We observe that longer history helps improve the model performance. (d) Comparison of different time-integration step length. A longer time-integration removes high-frequency information and simplifies the dynamics, making it easier to learn. On the other hand, the long-time property of particle diffusivity does not require the high-frequency information to estimate accurately. (e) Comparison of different connectivity radius when building the coarse-level graph. A radius of 6 Å is sufficient for modeling the SPE systems. (f) Comparison of different random seeds using the same model. Our model does stochastic rollouts and the performance is robust to random seeds.

**Fine-level Embedding GNN** $GN_E$. (Figure 7 (a)) Our model uses a fine-level embedding GNN to learn CG-bead embedding that enclose local structural information. We can remove this fine-level GNN and let the CG-bead embedding be the mean over the atom type embedding in each atom group. As shown in Figure 7 (a), model performance significantly drops without the Embedding GNN.

**Stochastic dynamics prediction.** (Figure 7 (a)) We attempt to let our model output deterministic acceleration at each forward simulation step. However, the inherent uncertainty makes the model predict very small movement for all particles at every step, and the model performance significantly drops. The small movements lead to slow transport of particles, and causes underestimation of Li-

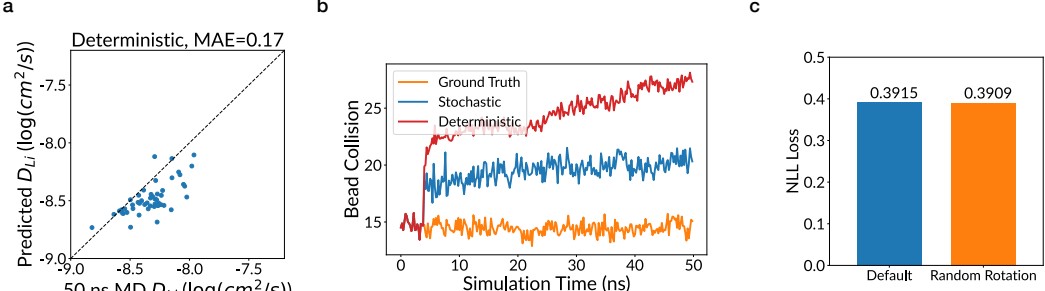

Figure 8: (a) Li-ion diffusivity estimation performance of the deterministic learned simulator. The model predicts small movements at every step, leading to slower ion transport and underestimate Li-ion diffusivity. (b) Bead collision as a function of time, averaged over the 50 testing SPEs. the deterministic model's prediction becomes increasingly unphysical as simulation proceeds. (c) The negative log-likelihood (NLL) loss (Equation (4)) of model prediction on testing polymers. Model performance remains the same when the input structures are randomly rotated before fed into the model.

ion diffusivity, as shown in Figure 8 (a). Such unrealistic dynamics also makes long simulation unstable. This is demonstrated in Figure 8 (b), which shows that deterministic model has higher bead collision with an increasing trend through time.

**Coarse-graining group size.** (Figure 7 (b)) We experimented with different coarse-graining group size. We observe that in terms of capturing particle diffusivity, using a group size of 7 outperforms finer (5) and coarser (15, 25) coarse-graining. We hypothesize that the finer system is hard to model accurately with limited training data, while the coarser system loses important information for accurate dynamics modeling. This result further suggests the optimal coarse-grained modeling should be conditional on the objective of MD simulation.

**Use of historical information for prediction.** (Figure 7 (c)) The spatio-temporal coarse-graining introduces memory effects to the resulting dynamics. Our model uses historical information by utilizing $k$-step historical velocities in dynamics prediction, and we experimented with $k = 0, 3, 8, 13, 18$, under a time-integration of 0.2 ns per step. We observe that longer history as input leads to better performance.

**Time-integration step size.** (Figure 7 (d)) We experimented with different time-integration step size. We observe that a longer time-integration gives the best performance. We hypothesize this is due to the dynamics simplification effect of long time-integration, while the loss of high-frequency dynamics does not damage estimation accuracy of particle diffusivity under long-time simulation.

**Connectivity radius.** (Figure 7 (e)) Our coarse-level graphs contain both CG-bonds and radius cut-off edges. The radius cut-off edges model non-bonded interactions. We experimented with radius 6, 7, 8, 9 Å, but found no significant performance difference. We conclude that a radius of 6 Å paired with geometric message passing is sufficient for modeling the SPE non-bonded dynamics.

**Random seeds.** (Figure 7 (f)) As our model does stochastic simulation, we examine its robustness against the random seed. We rollout 5 times using the same model and observe no significant performance difference across random seeds.

**Rotational equivariance.** Our model enforces translational invariance, but does not enforce rotational equivariance. Instead, our model is able to learn this equivariance from data and become "effectively rotational equivariant" when the equivariance exists (e.g., for the single-chain polymer system). We conduct experiments to apply random rotations at X, Y, and Z axes to input data, and compute the dynamics prediction loss, before and after the random rotations. Figure 8 (c) shows that the model test time performance is unchanged irrespective of the random rotations. Therefore, the time-integrated acceleration predicted by our model is accordingly rotated by the same amount as the input.

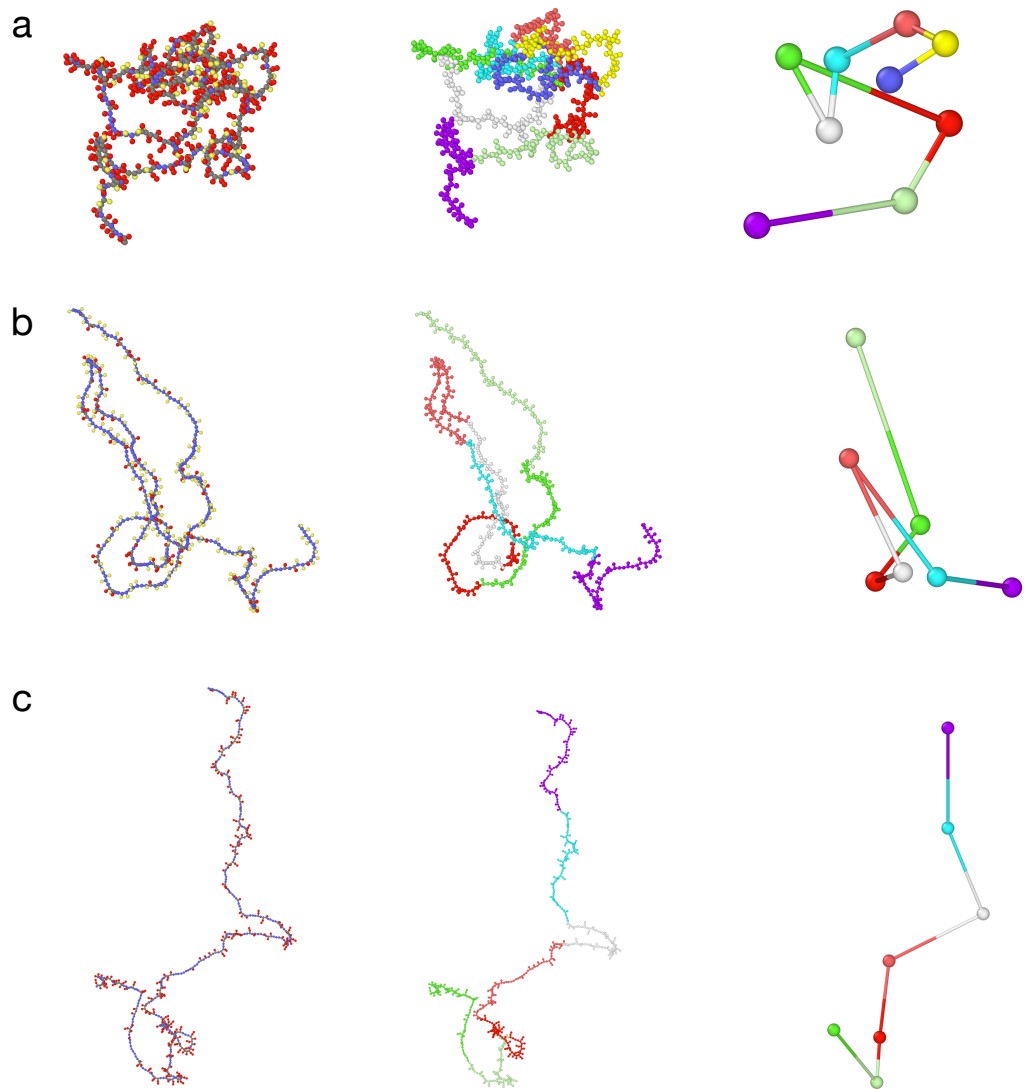

Figure 9: (a,b,c) The coarse-graining mapping of three example single-chain CG polymers. The first column is the original CG polymer structure with the color indicating the bead type; the second column colors beads according to their assigned super CG-bead; the third column shows the resulting CG system.

## F  COARSE-GRAINING ANALYSIS

We illustrate the coarse graining process for the single-chain CG polymers (Figure 9) and the SPE systems (Figure 10 (a)). For single-chain polymers, the graph clustering algorithm cuts the chian into segments of roughly equal sizes, and form a coarse-level chain, capturing the overall shape of the polymer. For SPE systems, atoms belonging to different molecules are never coarse-grained into the same group. So the Li-ions are never coarse-grained, while each TFSI-ion (with 15 atoms) is coarse-grained into two CG-beads. The system complexity is significantly reduced after coarse-graining. At a high level, the graph clustering algorithm groups nearby bonded atoms under the constraint that atom groups should be of similar sizes. Figure 10 (b,c) show the atom group size distribution for the single-chain polymer dataset (Figure 10 (b)) and the SPE dataset (Figure 10 (c)), confirming that the CG-beads are of similar sizes and contain the desired number of atoms.

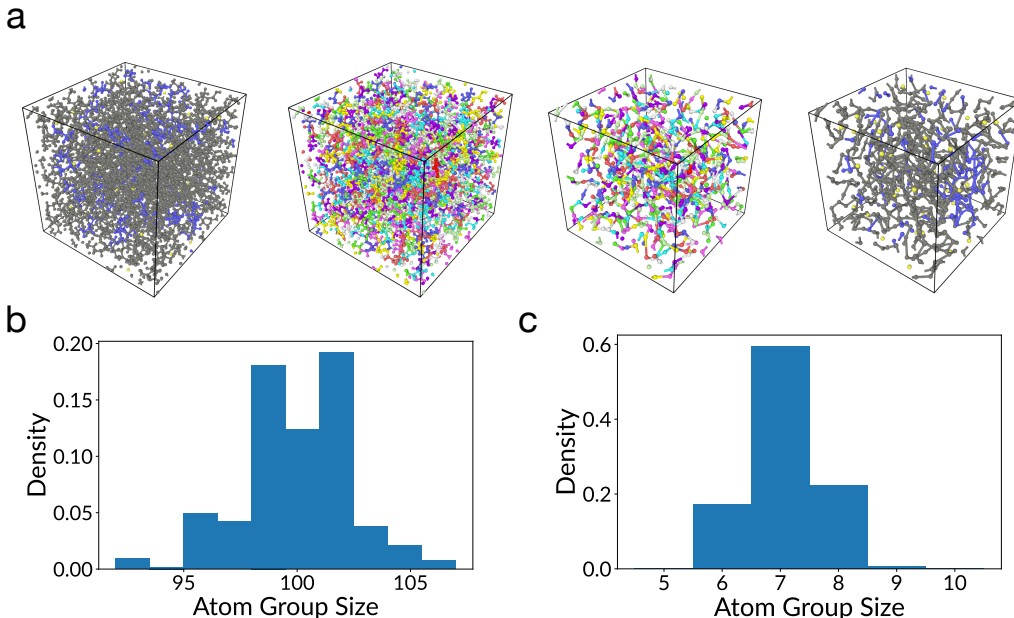

Figure 10: (a) The coarse-graining mapping an example SPE system. First column shows the full atomic structure; second column colors atoms according to their assigned CG-bead; third column shows the resulting CG system; fourth column shows the CG structure with color indicating the type of a bead: Li-ion, TFSI-ion or polymer. (b) Distribution of CG-bead atom group sizes for the single-chain CG polymer dataset. The clustering algorithm partitions a polymer into groups of roughly equal sizes of around 100. (c) Distribution of CG-bead atom group sizes for the SPE dataset. The clustering algorithm partitions the atoms into groups of roughly equal sizes of around 7.

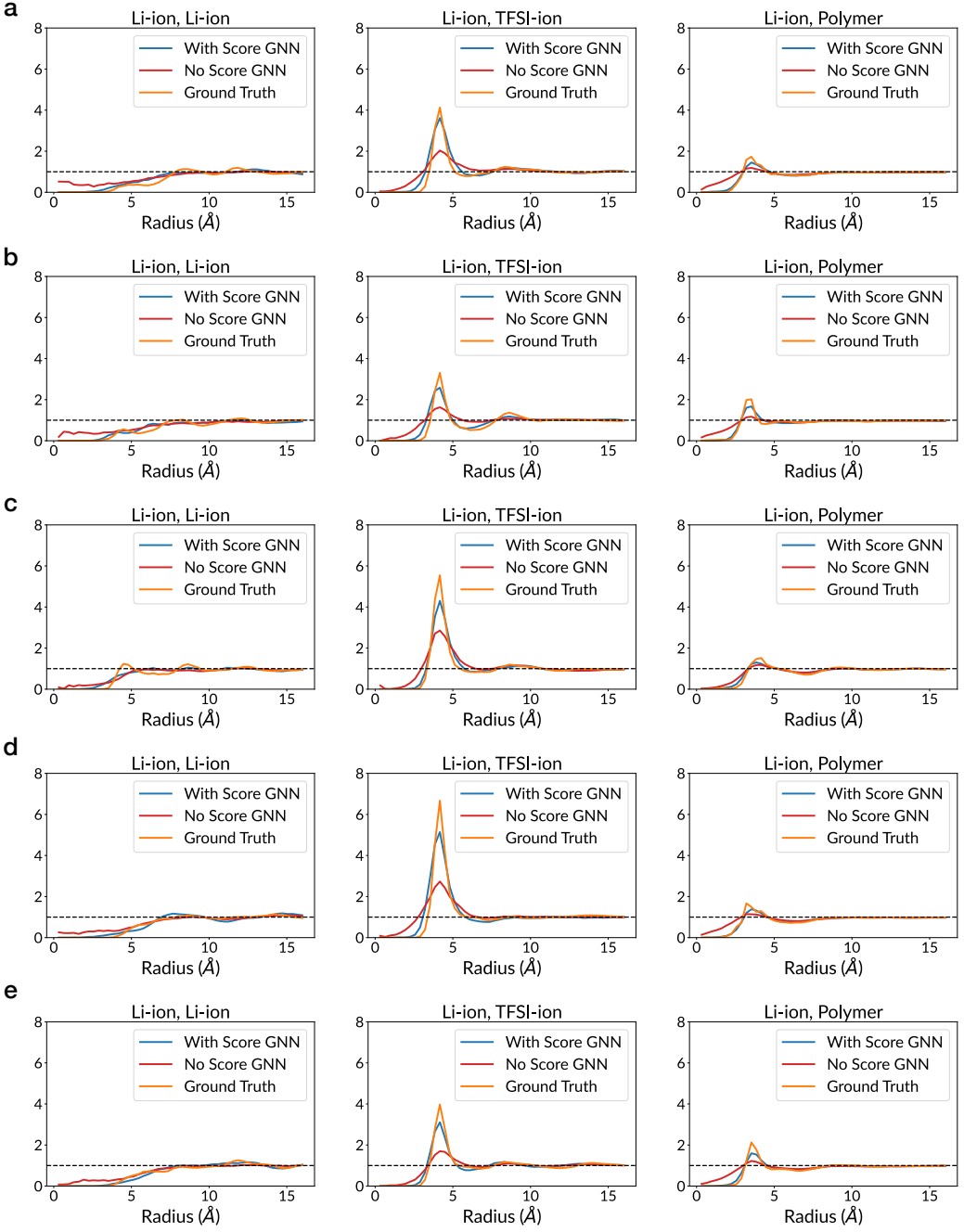

Figure 11: (a,b,c,d,e) Comparison of our model with/without the Score GNN refinement, and the ground truth MD simulation, on RDF of Li-ions (column 1), RDF of Li-ions and TFSI-ions (column 2), and RDF of Li-ions and polymer particles (column 3) for five sampled SPEs.

