# OpenReview forum: "Simulate Time-integrated Coarse-grained Molecular Dynamics with Geometric Machine Learning"
_ICLR.cc/2022/Workshop/DGM4HSD — ICLR 2022 DGM4HSD workshop Poster_

### Official Review · Reviewer_YHd5 · 2022-03-23
**The approach is impressive. Needs more experiments and external comparison.**

**Rating:** 6
**Confidence:** 4

**Review:**

**Summary and contributions:** Simulation of Molecular Dynamics (MD) is an important and challenging task. The paper is well-written and explains difficult concepts quite clearly. The authors propose the use of Graph Neural Networks along with a novel scoring algorithm to predict stable and long trajectories.

**Pros:** The method proposed by the authors is robust and significantly improves the time taken for simulating trajectories of Molecular Dynamics. The use of graph neural networks on the coarse-grained model is not a novel approach for this task. However, the score Graph Neural Network for providing stability in the case of long-time trajectories is commendable. The proposed method is tested against complex experimental settings and the ablation studies indeed prove the importance of each component of the method.

**Cons:** While it is clear that the experiments conducted to test the method's performance are complex, the important details about the experimental setup are missing. It is also not clear why the authors did not compare the performance (in terms of accuracy as well as time complexity) with the existing methods (like Noé et. al, 2019). Legend is missing in Figures 3(b) and 4(b).

**Quality & Significance:** The quality of this work is indeed high. The authors have used and well-explained the intuition behind the design of their geometric machine learning approach. The ability of the approach to accurately predict long trajectories just by training on small trajectories proves this and is an important step in making the MD simulators significantly faster without compromising on accuracy.

**Clarity & Originality:** The approach proposed in this work is novel as the use of Graph Neural Networks for scoring the predictions (in order to have stable outputs) is interesting and inspiring. While the authors explain their approach perfectly, more details about the experimental setup would be ideal. As mentioned earlier, Figures 3(b) and 4(b) do not have legends. At the same time, on page 3, Figure 2(g) is referred however Figure 2 has no subfigure g.

---

### Official Review · Reviewer_TFyx · 2022-03-25
**Simulate Time-integrated Coarse-grained Molecular Dynamics with Geometric Machine Learning**

**Rating:** 9
**Confidence:** 4

**Review:**

Goals:
The authors wish to establish a general-purpose model for efficient coarse-grained molecular dynamics simulations. We can split this goal into two sub-task: grouping atoms into beads and learning how to update bead positions given their previous position(s).
Description: The work describes an end-to-end learnable system composed of four components: atom embedding, coarse-graining, learning bead accelerations (dynamics), and refinement. The main methodological and conceptual innovations are in the last two steps --  these provide much-needed fresh ideas which are likely to impact the field more broadly. Specifically, the authors describe the prediction of stochastic accelerations rather than more commonly deterministic forces and then learn a denoising model to correct for imprecise bead displacements.

Evaluation:
The authors evaluate their model on two complex molecular systems and show impressive results. They show excellent predictions of equilibrium and dynamic expectation values, suggesting that they approximate well the coarse-grained dynamics and thermodynamics of the systems in question. Such an evaluation is appropriate for a Workshop paper as presented here. However, the authors should consider a more systematic generalizability/transferability analysis in later submissions.

Significance:
The paper describes a mature and new approach to coarse-grained MD using an end-to-end learnable system. This reviewer is confident that the work provides a significant conceptual advance to the field and provides ideas that may impact the generative modeling on highly structured manifolds more broadly. Highly suitable for presentation at this workshop.

Related Work and Discussion:
The authors acknowledge previous work in the field appropriately. Some discussions that could improve a future submission include the relationship of the presented method to variationally optimal coarse-graining schemes (force matching, relative entropy minimization, etc.). What are the possible limitations of the non-Markovian forward dynamics (k-step history dependence)? More systematically study the influence on the number of CG beads.

Clarity: The main manuscript is concise. Immediate questions were answered in the comprehensive appendix. Well done.
Recommendation: Accept.

---

### Official Review · Reviewer_kx1V · 2022-03-25
**Effective GNN-based CG model for long time MD simulation**

**Rating:** 7
**Confidence:** 3

**Review:**

This paper points out a critical problem that MD simulation is limited by the high computational cost and usually performs badly when predicting long-time properties. The authors propose a time-integration method to train the model on short MD trajectories and predict long-time properties, while graph clustering is used to construct a coarse-graining model and a refinement module is used to further refine the predicted structure. Single-chain polymer and multi-component Li-ion solid polymer electrolyte predictions are utilized to prove the effectiveness of the proposed method.

Pros:
1) This paper addresses an important problem in MD simulation, the time-integration approach successfully allows the model to learn from short-term trajectories and make predictions for long-time properties.
2) The paper is clearly written and easy to follow.
3) The experiments results seem impressive, and are well illustrated and discussed. The ablation studies validate the effectiveness of each component.

Cons:
1) There are only two baselines for the first experiment, GNN and LSTM, which seem too basic. And the settings of baseline models directly make the predictions without learning the dynamics seem not convincing since the input information is significantly decreased. Maybe the authors can explain in detail and include more comparison experiments.
2) The proposed method seems to work pretty good based on the significant improvement, but each component of the model is not novel.

---

### Decision · Program_Chairs · 2022-03-28

Accept (Poster)